# Acute Oral Toxicity and Genotoxicity Test and Evaluation of *Cinnamomum camphora* Seed Kernel Oil

**DOI:** 10.3390/foods12020293

**Published:** 2023-01-08

**Authors:** Pengbo Wang, Dongman Wan, Ting Peng, Yujing Yang, Xuefang Wen, Xianghui Yan, Jiaheng Xia, Qingwen Zhu, Ping Yu, Deming Gong, Zheling Zeng

**Affiliations:** 1State Key Laboratory of Food Science and Technology, Nanchang University, Nanchang 330047, China; 2Jiangxi Province Key Laboratory of Edible and Medicinal Resources Exploitation, Nanchang University, Nanchang 330031, China; 3School of Chemistry and Chemical Engineering, Nanchang University, Nanchang 330031, China; 4School of Food Science and Technology, Nanchang University, Nanchang 330031, China; 5Institute of Applied Chemistry, Jiangxi Academy of Sciences, Nanchang 330096, China; 6School of Resources and Environment, Nanchang University, Nanchang 330031, China; 7New Zealand Institute of Natural Medicine Research, 8 Ha Crescent, Auckland 2104, New Zealand

**Keywords:** *Cinnamomum camphora* seed kernel oil, acute oral toxicity, genotoxicity assessment, medium chain oil, safety assessment

## Abstract

*Cinnamomum camphora* seed kernel oil (CCSKO) is one of the important natural medium chain triglycerides (MCT) resources, with more than 95.00% of medium chain fatty acids found in the world, and has various physiological effects. However, CCSKO has not been generally recognized as a safe oil or new food resource yet. The acute oral toxicity test and a standard battery of genotoxicity tests (mammalian erythrocyte micronucleus test, Ames test, and in vitro mammalian cell TK gene mutation test) of CCSKO as a new edible plant oil were used in the study. The results of the acute oral toxicity test showed that CCSKO was preliminary non-toxic, with an LD_50_ value higher than 21.5 g/kg body weight. In the mammalian erythrocyte micronucleus test, there was no concentration-response relationship between the dose of CCSKO and micronucleus value in polychromatic erythrocytes compared to the negative control group. No genotoxicity was observed in the Ames test in the presence or absence of S9 at 5000 μg/mL. In vitro mammalian cell TK gene mutation test showed that CCSKO did not induce in vitro mammalian cell TK gene mutation in the presence or absence of S9 at 5000 μg/mL. These results indicated that CCSKO is a non-toxic natural medium-chain oil.

## 1. Introduction

Medium-chain triglycerides (MCT), composed of medium-chain fatty acids (MCFA) with six to twelve carbon atoms, can be completely hydrolyzed to free MCFA by tongue lipase, stomach lipase, and pancreatic lipase in vivo [1,2,3]. Free MCFA are mainly absorbed by small intestinal capillaries into the blood and then enter the liver via the portal vein [4]. Free MCFA can directly enter liver cells and mitochondria for oxidation and energy production or conversion into ketones without binding to carnitine. Most of the free MCFA are not converted into triglycerides or fats [5]. Therefore, the digestion of MCT is faster than that of long-chain triglycerides (LCT), and the absorption, transport, and metabolism of MCFA are faster than that of long-chain fatty acids (LCFA) [6,7,8].

Compared with LCFA, MCFA can reduce lipid accumulation by regulating key lipid-sensing genes [9], decrease the percentage of late apoptotic and necrotic cells by inhibiting the activities of caspase-3 and -9, ameliorate oxidative stress and inflammation by reducing the levels of inflammatory markers (IL-6, IL-1β and tumor necrosis factor-α) in human liver cells with steatosis [10,11]. Compared with LCT, MCT not only can supply energy rapidly [12] but also significantly decrease fat accumulation in adipose tissue, reduce body weight (BW) [13,14], improve blood and liver lipid profiles [15], enhance insulin sensitivity [5], alleviate glucose and lipid metabolic disorders in high fat diet-induced obese rats [16,17].

MCT have been used as cooking oil [18], cold cooking oil, oral and parenteral drug delivery medium [19,20], as well as ingredients in the production of energy food, slimming food, infant and young children formula food, infant and young children auxiliary food, sports nutrition food, MCT and MLCT fat emulsion [21,22]. MCT have a huge potential market for consumers.

There are very few kinds of oils rich in MCFA in the world. Currently, the commercially available sources of MCT are limited to coconut oil and palm kernel oil, of which MCFA account for 62% and 55% of total fatty acids, respectively [23,24]. Commercial MCT are synthesized from glycerol and medium-chain fatty acids that are obtained by hydrolyzing palm kernel oil and coconut oil and distilling mixed fatty acids [25].

The contents of caprylic acid, decanoic acid, and lauric acid in *Cinnamomum camphora* seed kernel oil (CCSKO) were 0.032–2.57%, 51.49–58.09%, and 35.00–40.08%, respectively [26]. CCSKO contains more than 95.00% MCFA and is one of the most important natural MCT found in the world. The CCSKO content in the *Cinnamomum camphora* seed kernel is more than 56%. The annual yield of *Cinnamomum camphora* seeds exceeds 15 million tons in China. Compared with lard and soybean oil, CCSKO was found to decrease body fat deposition and body weight and improve blood lipid profiles in healthy rats [27]. CCSKO also decreased fat accumulation in adipose tissue, reduced body weight, improved blood and liver lipid levels, increased insulin sensitivity, and improved dyslipidemia and lipid metabolic disorders in high-fat diet-induced obese rats [28,29]. However, to our knowledge, CCSKO has not been recognized as a safe oil or new food resource yet.

For derivatives or analogs to known substances or substances with a history of safe consumption in some countries and regions, the acute oral toxicity test and genotoxicity test can be performed to verify their oral toxicity (GB 15193.1-2014). There is a minor difference in composition between CCSKO and MCT, and in China’s Anfu County, CCSKO has been used as a cooking oil since the 1960s. The chemical composition and physicochemical properties of CCSKO need to be determined. In addition, the acute toxicity test and a standard battery of genotoxicity tests are required to confirm the edible safety of CCSKO for widespread use.

The study aimed to test and evaluate the acute oral toxicity and genotoxicity of CCSKO as a novel natural MCT oil. The results of this study may provide useful information and a basis for the subsequent nutraceutical and medicinal uses of CCSKO in humans. The discovery of the edible safety of CCSKO is equivalent to the edible safety discovery of tomatoes, called the Wolf peach, in the late 18th century.

## 2. Materials

### 2.1. Tested Substance

The test substance, CCSKO, was produced from the *Cinnamomum camphora* seeds (CCS) through the hot-pressing (HP) method [30] and an aqueous enzyme extraction (AEE) method [31], respectively. A detailed description of the methods is shown in Appendix A.

### 2.2. Main Reagents

The soybean oil was purchased in a local supermarket (Nanchang, Jiangxi Province, China). Calf serum was purchased from Beijing Solarbio Science & Technology Co., Ltd. (Beijing, China). Giemsa stain and phosphate buffer solution (PBS), dipotassium phosphate (K_2_HPO_4_·3H_2_O), and DMSO were purchased from Shanghai aladdin Biochemical Technology Co., Ltd. (Shanghai, China). D-biotin, L-histidine, ampicillin, and tetracycline were purchased from Avantor Inc. (Radnor, PA, USA). RPMI1640 medium was purchased from Thermofisher Scientific Inc. (Waltham, MA, USA). Methyl mesylate (MMS) and cyclophosphamide (CP) were purchased from Avantor Inc (Radnor, PA, USA). S9 (prepared in our laboratory), stored in the refrigerator at −80 °C for later use.

### 2.3. Animals and Housing Environment

The animal study has been approved by Jiangxi Provincial Center for Disease Control and Prevention, China. All the animals used were cared for according to the Guide for the Care and Use of Laboratory Animals, 8th edition, 2011 [32]. All procedures were approved by the Experimental Animal Ethics Committee, Nanchang University, China (Approval Code: No. 20211215-0114).

Fifty specific pathogen-free (SPF) Institute of Cancer Research (ICR) mice (half male and half female, 18–22 g BW) were provided by Hunan Shrek Jingda Laboratory Animal Co., Ltd. (Changsha, China) (License number: SCXK (Xiang) 2011–0003) for acute oral toxicity test, and another 50 SPF ICR mice (half male and half female, 25–30 g BW) were administrated for mammalian erythrocyte micronucleus test. The mice were housed in an animal room of Jiangxi Provincial Center for Disease Control and Prevention, China (Qualification certificate No. SYXK (Gan) 2012–0003) with a temperature of 21.2–24.6 °C and a relative humidity of 50–56%. The basic animal feed was provided by Beijing HFK Bioscience Co., Ltd (Beijing, China).

### 2.4. Strains and Cell Line and Culturing

#### 2.4.1. Strains for Bacterial Reverse Mutation Test (Ames Test)

Histidine deficient *Salmonella typhimurium* TA97 (CCC area +4 frameshift mutation), TA98 (CG area -1 frameshift mutation), TA100 (AT-GC base substitution, frameshift mutation partly), TA102 (GC-AT base substitution, frameshift mutation partly) and TA1535 (AT-GC base substitution, frameshift mutation partly) were provided by Jiangxi Provincial Center for Disease Control and Prevention, China.

These strains were cultured in a biosafety cabinet and CO_2_ incubator in the sterile room of Jiangxi Provincial Center for Disease Control and Prevention, China.

#### 2.4.2. Cell Line for In Vitro Mammalian Cell TK Gene Mutation Test

Mouse lymphoma cell line L5178Y TK^+/−^ clones (3.7.2C) were provided by Jiangxi Provincial Center for Disease Control and Prevention, China. These strains were cultured in a biosafety cabinet and CO_2_ incubator in the sterile room of Jiangxi Provincial Center for Disease Control and Prevention, China.

## 3. Methods

### 3.1. Determination of Chemical Composition, Thermal Behavior and Physicochemical Properties of CCSKO

The fatty acid composition, triglyceride composition, thermal behavior, and physicochemical properties analysis of CCSKO were determined. A detailed description of methods is shown in Appendix A.

### 3.2. Acute Oral Toxicology Test of CCSKO

#### 3.2.1. Test Samples Processing

Four CCSKO samples (21.5 g, 20.0 g, 9.28 g, and 4.30 g) were diluted to 40 mL with soybean oil (SO), respectively. These solutions were used as the test substances for gavage at a dose of 0.4 mL/20 g BW.

#### 3.2.2. Animals Feeding and Test

After 5-days of environmental adaptation and quarantine observation under experimental conditions, 50 animals (25 males and 25 females) were selected and randomly divided into 5 groups (*n* = 10, 5 males and 5 females). Females were nulliparous and non-pregnant.

All the mice were fasted overnight. The CCSKO gavage doses of mice were 21.5, 10.0, 4.64, and 2.15 g/kg BW, respectively. An extra group of mice was administrated soybean oil at the dose of 21.5 g/kg BW as a negative control. The mice in the 21.5 g/40 mL groups (both CCSKO and soybean oil groups) were given gavage twice a day, with an interval of 4 h, and the other groups were given gavage once a day. These mice were treated with CCSKO or soybean oil, and their poisoning symptoms and death were observed and recorded continuously for 14 days. During the experiment, these mice were fed a basic diet and drank freely.

At the end of the observation period, all the mice were weighed, and general clinical signs were observed. The mice were then sacrificed by isoflurane inhalation, and all organs were visually examined following animal autopsies.

### 3.3. Mammalian Erythrocyte Micronucleus Test of CCSKO

#### 3.3.1. Test Samples Processing

The CCSKO was weighed as the test substances in the high-dose group, and the test substances in the medium-dose and low-dose groups were prepared by half-dilution of the test substances in the high-dose group and the medium-dose group with SO, and these solutions were gavaged with the capacity of 4 mL/kg BW. The density of the CCSKO sample was 0.86 g/mL. Therefore, the high, medium, and low dose groups were 3.44 g, 1.72 g, and 0.86 g/kg BW, respectively.

#### 3.3.2. Animals Feeding and Test

After 5-days of environmental adaptation and quarantine observation under experimental conditions, the 50 mice were divided into 5 groups, with 10 mice in each group, half male and half female.

The test was carried out by oral gavage for 30 h. All of the mice were fasted overnight. Three dose groups of CCSKO were set at 3.44 g/kg BW, 1.72 g/kg BW, and 0.86 g/kg BW. Cyclophosphamide (CP) (40 mg/kg BW) was used as a positive control, and soybean oil (SO) was used as a negative control. CCSKO solution was administered twice at 24-h intervals. Six hours after the last gavage, the mice were sacrificed by isoflurane inhalation, and the femoral bone marrow was collected and smeared with calf serum dilution, fixed with methanol, and stained with Giemsa. Under an optical microscope, 2000 polychromatic erythrocytes (PCE) were counted from each mouse to observe the number of polychromatic erythrocytes containing micronucleus to calculate the PCE/normochromatic erythrocytes (NCE) value, and the number of mature erythrocytes in the field of 200 PCE were observed to calculate the micronucleus rate. The PCE/ NCE value and micronucleus rate were statistically analyzed by χ^2^ test Fisher exact probability test.

### 3.4. Bacterial Reverse Mutation Test (Ames Test) of CCSKO

#### 3.4.1. Test Samples Processing

One gram of the CCSKO samples was weighed and dissolved in DMSO to 20 mL, which was the highest concentration. This solution was autoclaved for 20 min at a pressure of 0.103 MPa. For the preliminary test, 2 mL of the sample solution of this concentration was taken, and 4.32 mL of sterile DMSO was added and mixed evenly. In this way, the last concentration of the sample solution was diluted $\sqrt{10}$ (about 3.16) times to the fifth concentration one after another, and the corresponding experimental doses were 5000, 1582, 501, 158.5, and 50.2 μg per petri dish, respectively. For the main test, 1 mL of this concentration sample was taken, and 4 mL of sterile DMSO was added and mixed evenly. In this way, the last concentration of sample solution was diluted 5 times to the fifth concentration in sequence, and the corresponding experimental doses were 5000, 1000, 200, 40, and 8 μg per petri dish, respectively.

#### 3.4.2. Strains Culturing and Test

Histidine-deficient *Salmonella typhimurium* TA97, TA98, TA100, TA102, and TA1535 test strains were used in this study. Polychlorinated biphenyl (PCB)-induced rat liver homogenate was used as an in vitro metabolic activation system (S9). Five doses of 5000, 1582, 501, 158.5, and 50.2 μg/dish were set up in the preliminary experiment, and five doses of 5000, 1000, 200, 40, and 8 μg/dish in the main experiment, and blank control (untreated control) was set up. At the same time, solvent control (DMSO) and positive control (NaN_3_, 2-AF, Dexon, 1,8-DHAQ, and CP) were set up.

The 0.1 mL of test strain enrichment solution, 0.1 mL of test substance solution, and 0.5 mL S9 mixture (when metabolic activation was required) were added to the top layer of agar medium, mixed, and poured onto the bottom medium plate. Three parallel plates were made for each dose group. The test strains were incubated at 37 °C for 48 h, and the number of revertant colonies per dish was counted.

### 3.5. In Vitro Mammalian Cell TK Gene Mutation Test of CCSKO

#### 3.5.1. Test Samples Processing

The 1 g of the sample was weighed and dissolved in DMSO to make a 2 mL solution. The solution was filtered, sterilized, and successively diluted with DMSO into two concentration solutions. Three concentration solutions were used as high, medium, and low dose groups. Since the additional amount was 1% of the volume, the test doses were 5, 2.5, and 1.25 mg/mL, respectively. At the same time, a negative control group (H_2_O), positive control group (−S9: methyl methylsulfonate (MMS), +S9: cyclophosphamide (CP)) and solvent (DMSO) control group were set up. The experiment was carried out under the condition of adding and not adding S9.

#### 3.5.2. Cell Lines Culturing and Test

The mouse lymphoma cell lines L5178Y TK+/−clone (3.7.2C) were treated in THMG (T, thymidine; H, hypoxanthine; M, methotrexate; G, glycine) medium for 24 h before the assay. Then the cell lines by centrifugation and rinsing were cultured in THG medium for 48 h. The sediment was cultured in the RPMI1640 medium containing 10% horse serum under 5% CO_2_, 37 °C, and saturated humidity. The well-grown cells were taken, and the cell density was adjusted to 5 × 10^5^ /mL. The process mentioned above was repeated to adjust the cell density to 2 × 10^5^ /mL.

#### 3.5.3. PE_0_ (Plate Inoculation Efficiency at Day 0) Determination

An appropriate amount of cell suspension was diluted to 8 cells /mL and inoculated into 96-well plates (200 μL was added to each well, that is, 1.6 cells per well on average). One plate was made for each dose and incubated at 37 °C, 5% CO_2_, and saturated humidity for 12 days.

#### 3.5.4. PE_2_ (Plate Inoculation Efficiency at Day 2) Determination

At the end of the 2-day expression culture, an appropriate amount of cell suspension was taken, diluted in a gradient according to Section 3.5.3, and seeded in 96-well plates. After 12 days of culture, the number of wells with colony growth in each plate was counted.

#### 3.5.5. TFT Resistance Mutation Frequency (tk-MF) Determination

After 2 days of expression culture, an appropriate amount of cell suspension was taken, the cell density was adjusted to 1 × 10^4^ cells /mL, and TFT (trifluorothymidine, final concentration was 3 μg/mL) was added, mixed, and inoculated into 96-well plates. A volume of 200 μL was added to each well, that was, 2000 cells per well. Two parallel experiments were performed for each dose group, and the plates were incubated in an incubator with 5% CO_2_ at 37 °C and saturated humidity for 12 days. The number of colony growth wells in each plate was counted by the naked eye.

#### 3.5.6. Data Processing

(1)Plate efficiency (PE_0_ and PE_2_)
$$\mathrm{PE}=-\frac{1}{1.6}\ln\frac{\mathrm{EW}}{\mathrm{TW}}\times 100\%$$
where EW is the number of empty wells, TW is the number of total wells, and 1.6 is the number of cells inoculated per well.(2)Relative suspension growth (RSG)
$$\mathrm{RSG}=\frac{\mathrm{CMT}}{\mathrm{CMN}\;\mathrm{or}\;\mathrm{CMM}}\times 100\%$$
where CMT is cell multiplication in the treatment group during expression, CMN is cell multiplication in a negative group during expression, and CMM is cell multiplication in the menstruum group during expression.(3)Relative survival (RS)
$$\mathrm{RS}=\frac{\mathrm{PE}\;(\mathrm{processed})}{\mathrm{PE}\;(\mathrm{control}/\mathrm{menstruum})}\times 100\%$$(4)TFT resistance mutation frequency (MF)
$$\mathrm{MF}\left({\times 10}^{-6}\right)=-\frac{\ln\left(\mathrm{EW}/\mathrm{TW}\right)}{N}{/\mathrm{PE}}_{2}$$
where EW is the number of empty wells, TW is the number of total wells, N is the number of cells inoculated per well (the number of L5178Y was 2000), PE_2_ is the plate efficiency in day 2.(5)Relative total growth (RTG)
RTG=RSG×RSn×100%RSn, relative survival of L5178Y cell in the second day (RS2).(6)Small colony mutation or slowly-growth colony mutation (SCM)
$$\mathrm{SCM}=\frac{S\text{-}\mathrm{MF}}{T\text{-}\mathrm{MF}}\times 100\%$$S-MF is the mutation frequency of a small colony or slowly-growth colony; T-MF is the total mutation frequency.

### 3.6. Statistical Analysis

Results were expressed as mean ± standard deviation (SD). Data were analyzed by one-way ANOVA with SPSS 26.0 software (SPSS, Inc., Chicago, IL, USA). Differences with a *p* < 0.05 were considered statistically significant.

## 4. Results and Discussion

### 4.1. Chemical Composition and Physicochemical Properties of CCSKO

#### 4.1.1. Fatty Acid Composition of CCSKO

As shown in Table 1, a total of nine fatty acids were identified in CCSKO obtained by HP and AEE method. It was worth mentioning that CCSKO had the highest content of caprylic acid (C10:0, 61.18 ± 0.94 and 58.09 ± 0.95%), followed by capric acid (C12:0, 35.94 ± 0.73 and 37.80 ± 0.74%). This result indicated that the fatty acid in CCSKO was mainly composed of MCFA, which comprised more than 95% of the content. The MCFA content in CCSKO was significantly higher than that of palm kernel oil (55%) [23] and coconut oil (62%) [24], which could be regarded as a natural medium-chain oil. Additionally, CCSKO contained a certain amount of unsaturated fatty acids, including oleic acid (C18:1, 1.18 ± 0.03 and 1.81 ± 0.04%), linoleic acid (C18:2, 0.32 ± 0.00 and 0.39 ± 0.01%) and linolenic acid (C18:3n-3, 0 and 0.09 ± 0.01%). The fatty acid composition of *sn*-2 and *sn*-1.3 in CCSKO was similar to the total fatty acid composition.

#### 4.1.2. Triglyceride Composition of CCSKO

Three main peaks in CCSKO obtained by HP and AEE were observed (Appendix A), suggesting that CCSKO was mainly composed of three types of triglycerides. Based on the results of fatty acid composition, the equivalent of carbon number (ECN) was used to indicate the triglyceride structure of CCSKO. ECN was calculated by subtracting the number of carbon atoms in the acyl residue of triglyceride from the number of double bonds of fatty acids forming triglyceride [33]. The retention time of triglyceride was linearly related to ECN and the number of double bonds of triglyceride in the elution [34]. As shown in Table 2, CCSKO had the highest content of C-C-La (ECN = 32, 84.78 ± 3.12% and 84.81 ± 3.59%) type of triglyceride, followed by C-La-La (ECN = 34, 9.82 ± 0.30 and 9.71 ± 0.33%) and C-C-C (ECN = 30, 5.40 ± 0.09% and 5.38 ± 0.09%). These results revealed that CCSKO was an MCT.

Considering the ECN of CCSKO obtained by the two methods were equivalent, only the total ion chromatogram (TIC) of CCSKO obtained by the AEE method was shown. The TIC (Figure 1A and Table 2) of CCSKO was similar to the HPLC profile. For the coupling of HPLC and MS analysis, both protonated molecules [M + H]^+^ and fragment ions [M + H − RCOOH]^+^ were generated, which was important for elucidating the triglyceride structure of CCSKO. As shown in Figure 1B, peak 1 contained the molecular ion at m/z 572.4 [M + H]^+^ with a fragment at m/z 383.3 [M + H − RCOOH]^+^, indicating the loss of capric acid. Peak 2 (Figure 1C) with molecular ion at m/z 600.5 [M + H]^+^ and fragments at m/z 411.3 and 383.3 [M + H − RCOOH]^+^ represented the loss of capric acid and lauric acid, respectively. Peak 3 (Figure 1D) had molecular ion at m/z 628.5 [M + H]^+^, and produced fragments at m/z 439.3, 411.3, 383.3 [M + H − RCOOH]^+^. Combined with the results of retention time and ECN of triglyceride in CCSKO analyzed by HPLC-ELSD, peaks 1, 2, and 3 were identified as C-C-C, C-C-La, and C-La-La, respectively. This result was consistent with the HPLC-ELSD analysis, which further confirmed that CCSKO was a natural source of MCT.

#### 4.1.3. Thermal Behavior of CCSKO

Differential scanning calorimetry (DSC) was used to evaluate the thermal stability of CCSKO. DSC can reflect the information on phase transition during the melting and crystallization of the sample. As shown in Figure 2A, CCSKO obtained from HP and AEE had a major melting peak (endothermic peak) with −17.67–8.55 °C and −15.31–−6 °C, respectively, and the peak melting temperatures were −9.54 °C and −9.86 °C, respectively. The shoulder melting peak with −18.45–7.47 °C and −19.33–2.46°C represented the melting temperature of unstable crystals of the low-melting TAG species that prematurely melted [35]. Moreover, CCSKO had a crystal peak (exothermic peak) of 5.36–23.42 °C and 8.05–25.89 °C, respectively, and the peak crystal temperatures were 19.16 °C and 20.86 °C, respectively. This phenomenon was caused by the characteristics of the main triglyceride structure of CCSKO, including C-C-La, C-La-La, and C-C-C. It was reported that coconut oil had two crystal peaks at −4.43 and 1.02 °C [36]. Based on the differential chemical composition between coconut oil and CCSKO, the result of DSC analysis also indicated that the fatty acid composition and triglyceride structure of CCSKO were relatively simpler. The low melting temperature of CCSKO ensures a smooth liquid state at room temperature. Moreover, the crystallization temperature of CCSKO is lower than human body temperature (37 °C), which can provide the basis for its use as edible oil and functional, structural lipids.

#### 4.1.4. FTIR Spectrum of CCSKO

The FTIR spectrum of CCSKO is shown in Figure 2B. It is worth noticing that the FTIR spectrum of CCSKO obtained by the HP and the AEE methods were identical. The functional groups of CCSKO responsible for FTIR absorption were as follows: the peaks near 2920 cm^−1^ and 2850 cm^−1^ relating to the stretching vibration of CH_3_ and CH_2_, respectively, the peak near 1740 cm^−1^ relating to the ester bond (C=O), the peak near 1464 cm^−1^ relating to the bending vibration of CH_2_, the peak of 1158 cm^−1^ corresponding to the stretching vibration of −C-O and CH2 bending, and the peak of 1097 cm−1 corresponding to the bending and vibration peak of –C-O [37,38]. In addition, the functional groups of CCSKO were similar to those of most plant oils, such as coconut oil [39] and palm kernel oil [40], mainly due to the similar triglyceride structure of these oils.

#### 4.1.5. Physicochemical Properties of CCSKO

The CCSKO that was in line with the requirements of a food grade standard was used for further experiments. As shown in Table 3, the levels of moisture and volatile matter, insoluble impurity, acid value, and peroxide value of CCSKO obtained by the HP and the AEE method were determined to be 0.17% and 0.13%, 0.0059%, and 0.0063%, 0.88 mg KOH/g, and 0.40 mg KOH/g and 0.02 mmol/kg and 0 mmol/kg, respectively. In comparison with the Agricultural Industry Standard of China (NY/T 230−2006, coconut oil), the contents of moisture, volatile matter, and acid values of CCSKO were slightly higher, while the insoluble impurity and peroxide values of CCSKO were significantly lower. In comparison with the National Standards of China (GB/T 230−2006, palm kernel oil), the contents of moisture, volatile matter, insoluble impurity, and peroxide values of CCSKO meet the physicochemical standards of refined palm kernel oil. Additionally, the iodine values of CCSKO were 5.7 g I_2_/100 g and 3.9 g I_2_/100 g, which were significantly lower than the standards for coconut oil and palm kernel oil. No precipitates, Roverbon colorimetric red value change, and increased yellow value by 1.0 were observed in the heating test. The contents of moisture and volatile matter, insoluble impurity, and unsaponifiable matter content of transparent, faint yellow CCSKO without odor were 0.17% and 0.13%, 0.0059% and 0.0063% and 0.34% and 0.35%, respectively, which verified the purity of CCSKO. No arsenic, plumbum, benzopyrene (α), or aflatoxins B1 was detected in CCSKO.

Considering that the physicochemical properties (moisture and volatile matter, acid value, iodine value, and peroxide value) of CCSKO obtained by the AEE method were better than that by the HP method, the CCSKO obtained by the AEE method was used for subsequent tests.

### 4.2. The Acute Oral Toxic Effects of CCSKO in SPF ICR Mouse

The acute oral toxicity test can provide information on the health hazards caused by oral exposure to test substances in a short period of time, thus providing the basis for the dose selection of further toxicity tests and preliminarily estimates of the target organs and possible mechanisms of toxicity [41].

According to the results above, the purity of CCSKO obtained by the HP method was 99.51%, indicating that the CCSKO can be used in acute oral toxicology tests. As shown in Table 4, none of the SPF ICR mice fed with CCSKO at 2.15–21.5 g/kg BW or SO at 21.5 g/kg BW showed toxic symptoms or death during the consecutive 14-day experiment. BW was measured before and after CCSKO administration, respectively. There was no significant difference (*p* > 0.05) in the final BW among different groups. No animals died during the dosing period, and no macroscopic lesions were observed in gross anatomy (gross dissection) in all the mice (data not shown).

Based on these findings, the acute oral toxicity LD_50_ of CCSKO to female and male mice was determined to be more than 21.5 g/kg BW, which was greater than that of medium and long chain triglyceride (MLCT) (over 5 g/kg BW) previously reported by Matulka et al. [42]. According to the standard of acute toxicity dose classification, CCSKO was actually non-toxic [41]. Therefore, acute oral toxicity evaluation showed that CCSKO was preliminary safe for consumption.

### 4.3. The Genotoxicity Effects of CCSKO in SPF ICR Mouse

Generally, a single genotoxicity test implemented using individual endpoints cannot determine all genotoxicity aspects. A standard battery testing approach is feasible since no single test is capable of detecting all genotoxic mechanisms [43]. Assessment of genotoxicity is generally the most common and important, as botanical components are used at levels above what is generally considered the highest level in plant toxicology [44]. Several genotoxicity assessments of MCT have been conducted previously [42,45]. Given the physicochemical properties of CCSKO, it is important to evaluate its mutagenicity and genotoxicity. The standard battery of genotoxicity tests were conducted under the guidance of the International Council for Harmonization of Technical Requirements for Pharmaceuticals for Human Use (ICH) [46]. The potential genotoxicity was evaluated using the mammalian erythrocyte micronucleus test, Ames test, and in vitro mammalian cell TK gene mutation analysis. In conclusion, a standard battery of in vitro and in vivo genotoxicity assays demonstrated no genotoxicity for CCSKO.

#### 4.3.1. Micronucleus Rate Change in Mammalian Erythrocyte Micronucleus Test of Mice

Generally, the results were considered positive when the micronucleus rates showed a significant concentration-response relationship and the micronucleus rate in the negative control group was greater than 5‰ [47]. As shown in Table 5, the micronucleus rates in the male and female mice were less than 5‰ (1.20 ± 0.27‰ and 1.10 ± 0.42‰, respectively), and the PCE/ NCE values in all dose groups were not less than 20% of that in the solvent control group, indicating that the results were valid. In addition, the increase in micronucleus was within the 95% confidence limits of the solvent control, was not CCSKO dose-related, and no increases in micronucleus frequency were considered incidental, while the rate of micronucleus in the cyclophosphamide (CP) positive control group was obviously higher than the solvent control group. In summary, CCSKO did not induce micronuclei when incubated with a mouse marrow red blood cell (RBC) system with or without metabolic activation. The result indicated that CCSKO did not have a mutagenic effect.

The mammalian erythrocyte micronucleus test, the most frequently used test in the list of in vivo genotoxicity tests [48], was used to determine the genotoxicity of CCSKO in mouse PCEs [49,50]. No apparent concentration-response relationships were observed in the presence or absence of metabolic activation, suggesting that CCSKO did not have significant cytotoxicity or mutagenicity. Therefore, the test result was considered negative, indicating that CCSKO did not induce an increase in mammalian PCEs. Further, the micronucleus rates in sex-specific mice from all doses of CCSKO were not significantly different from that of the control, indicating that CCSKO did not have a mutagenic effect. Interestingly, the micronucleus responses in male and female mice have been reported to be similar [51]. Therefore, most of the tests were conducted either in both sexes or in either sex. Extra factorial design can thus be used if mice with both sexes are used to make full use of experimental resources in the future.

#### 4.3.2. Change in the Number of CCSKO- Induced Reverse Mutation Colonies in Bacterial Reverse Mutation Test (Ames Test)

A result can be considered positive when the number of CCSKO-induced reverse mutation colonies is not less than twice that of the untreated group and in line with one of the following two conditions: (a) a dose-reaction relationship is present, (b) a repeatable positive result is available at one test point [52]. It was found that CCSKO did not increase the number of mutant colonies in the test strains TA97, TA98, TA100, TA102, or TA1535 in the preliminary and main tests with or without metabolic activation (Figure 3 and Table 6). (Considering the inconvenience of presenting the numbers of reverse bacteria colonies in the positive group and menstruum group, and the purpose was to determine whether there was an obvious dose-reaction relationship, the numbers of reverse bacteria colonies in the positive and menstruum groups were not shown Figure 3). Conversely, the positive control substances showed a significant mutation in the respective test strains.

The Ames test was conducted to determine the mutation behavior of the test strains when changing from a specific amino acid deficiency (i.e., histidine) state to a normal strain. In this test, the number of all tested strains was not increased in the presence or absence of an S9 mixture, and the results did not in line with the positive criteria. Therefore, the results could be considered negative. The genotoxicity of MCT was assessed by Traul et al. [45], who found a weak to non-genotoxic capability of MCT, which was consistent with the results presented herein. Interestingly, the number of bacteria colonies induced by different doses of CCSKO was mostly lower than those in the untreated control and the menstruum control group, indicating the antibacterial capacity of CCSKO. The result was similar to a previous study showing that monocaprin, the monoglyceride of one of the most important substances (i.e., decanoic) in CCSKO, effectively sterilized *E. coli* in juice without causing significant damage to quality [53], which may provide a good idea for the future manufacture of beverages or nutrient solutions composed of CCSKO. On the other hand, we can further investigate whether CCSKO may interfere with the mammalian cell replication system through mammalian cell mutation experiments.

#### 4.3.3. Change in the Spontaneous Mutation Frequencies of In Vitro Mammalian Cell TK Gene in In Vitro Mammalian Cell TK Gene Mutation Test

The test results can be considered valid when the spontaneous mutation frequencies are in the range of 50 × 10^−6^ − 200 × 10^−6^, and the PE_0_ and PE_2_ values are in the range of 60–140% and 70–130%, respectively [54]. In addition, the total colony mutation frequency (T-MF) of the positive control group should be: (a) significantly different from or greater than three times that of the negative/menstruum control group, (b) increased mutation frequencies were observed with relative survival (RS) rate below 20%. As shown in Figure 4, Appendix A, the total colonies mutation frequency (T-MF) of the negative control group was within the specified range in the presence and absence of metabolic activation (87.47 ± 4.95 ×10^−6^ and 54.36 ± 8.12 ×10^−6^), and PE_0_ (89.30% and 98.04%) and PE_2_ (86.64% and 101.24%) values were also within the specified range, indicating that the results were valid. Furthermore, all doses of CCSKO showed T-MF values no 3-fold more than those of the negative/menstruum control group in the presence and absence of S9, and there was no concentration-response relationship. Therefore, there was no mutagenic toxicity in CCSKO.

There was no mutagenic toxicity in CCSKO in the presence or absence of S9. However, this experiment required an exogenous source of metabolic activation (S9 was used in this test). Thus, the in vitro test may not completely simulate metabolic conditions in mammals. Therefore, the results of this experiment cannot be directly extrapolated to mammals. In addition, consideration should be given to avoiding conditions (including potential interaction with the test system) that could lead to positive artifactual results [54].

## 5. Conclusions

In this study, the physicochemical profiles, an in vivo acute oral toxicity test, and a battery of in vitro genotoxicity tests were performed to evaluate the safety of using CCSKO as a novel natural MCT. For the physicochemical profile determination of CCSKO, it was one of the important natural sources of MCT with good quality and low content of unsaturated fatty acids. For the acute oral toxicity test, no obvious poisoning symptoms or death of the mice fed with CCSKO with all doses (21.5, 10.0 4.64, 2.15 g/kg BW) were observed during the observation period of 14 days. Therefore, the LD_50_ of CCSKO in female and male mice was more than 21.5 g/kg BW. For a battery of genotoxicity tests, compared to the negative control, no significant genotoxicity was observed in the mammalian erythrocyte micronucleus test, Ames test, or in vitro mammalian cell TK gene mutation test. In consequence, the results of this study showed that CCSKO presents no acute oral toxicity in mice and does not cause genotoxicity, which has a huge potential market for consumers. However, in addition to the preliminary safety data and theoretical foundation obtained in this study, further long-term oral toxicity of CCSKO should be performed to satisfy the overall safety requirements for the verification of health-functional foods.

## Figures and Tables

**Figure 1 foods-12-00293-f001:**
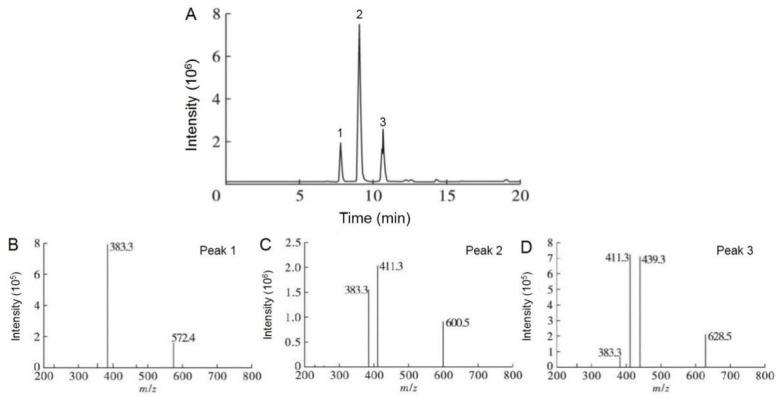
Total ion chromatogram (**A**) and MS/MS spectra (**B**–**D**) of CCSKO.

**Figure 2 foods-12-00293-f002:**
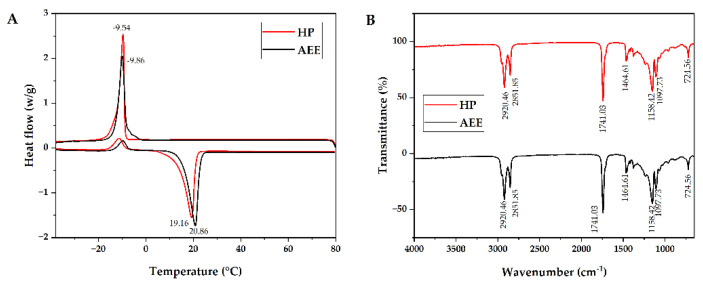
DSC curve (**A**) and FTIR spectrum (**B**) of CCSKO.

**Figure 3 foods-12-00293-f003:**
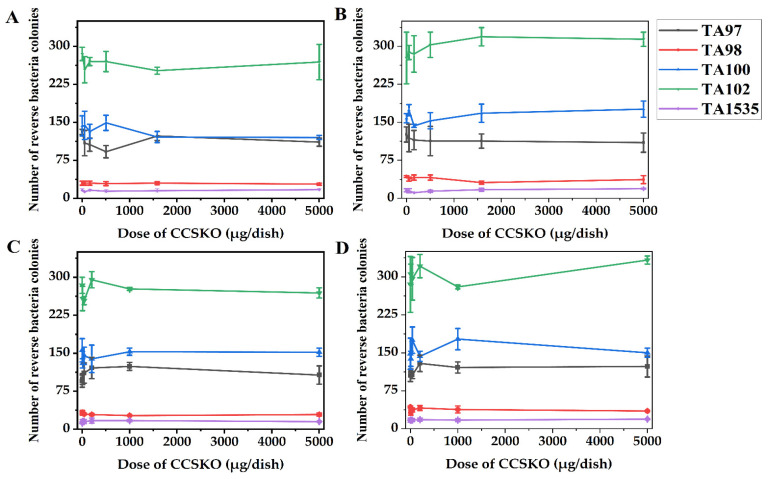
The relationship between the dose of CCSKO (μg/dish) and the number of reverse bacteria colonies in the experimental group. (**A**) The preliminary test in the absence of S9 (−S9). (**B**) The preliminary test in the presence of S9 (+S9). (**C**) The main test in the absence of S9 (−S9). (**D**) The main test in the presence of S9 (+S9). Data are expressed as mean ± SD.

**Figure 4 foods-12-00293-f004:**
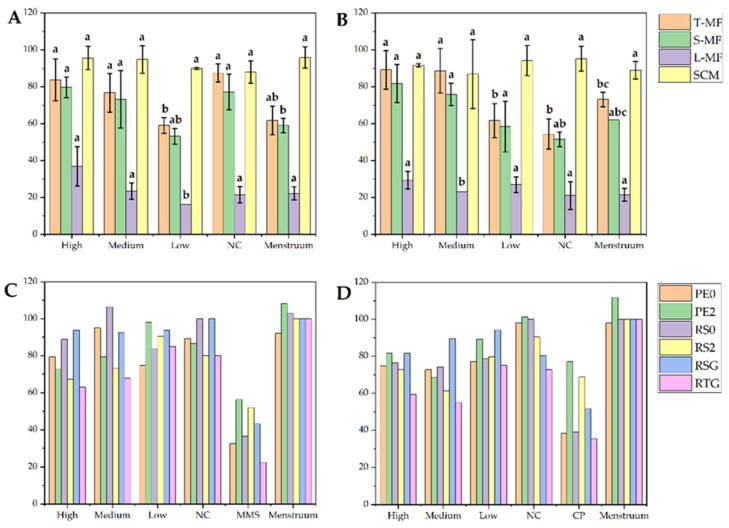
L5178Y cell TK gene mutation frequency and cell toxicity of in vitro mammalian cell TK gene mutation test of CCSKO. (**A**) L5178Y cell TK gene mutation frequency in the absence of S9 (−S9) induced by CCSKO. (**B**) L5178Y cell TK gene mutation frequency in the presence of S9 (+S9) induced by CCSKO. (**C**) The cell toxicity of in vitro mammalian cell TK gene mutation test of CCSKO in the absence of S9 (−S9). (**D**) The cell toxicity of in vitro mammalian cell TK gene mutation test of CCSKO in the presence of S9 (+S9). NC, negative control. Data in Figure 4A,B are expressed as mean ± SD; Values with different letters in the same color indicate significant differences (*p* < 0.05).

**Table 1 foods-12-00293-t001:** Fatty acid composition of CCSKO.

Type of Fatty Acid	Fatty Acid Composition (%)					
	HP			AEE		
	Total	*sn*-2	*sn*-1,3	Total	*sn*-2	*sn*-1,3
C8:0	0.43 ± 0.03 ^a^	0.50 ± 0.02 ^a^	0.39 ± 0.01 ^a^	0.44 ± 0.03 ^a^	0.47 ± 0.03 ^a^	0.43 ± 0.02 ^b^
C10:0	61.18 ± 0.94 ^a^	58.68 ± 0.79 ^a^	62.43 ± 0.55 ^a^	58.09 ± 0.95 ^b^	57.96 ± 0.80 ^a^	58.16 ± 0.56 ^b^
C12:0	35.94 ± 0.73 ^a^	37.21 ± 0.59 ^a^	35.31 ± 0.36 ^a^	37.80 ± 0.74 ^b^	38.01 ± 0.60 ^a^	37.7 ± 0.37 ^b^
C14:0	0.99 ± 0.03 ^a^	1.02 ± 0.03 ^a^	0.65 ± 0.02 ^a^	0.99 ± 0.03 ^a^	1.03 ± 0.04 ^a^	0.97 ± 0.03 ^b^
C16:0	0.18 ± 0.00 ^a^	0.14 ± 0.01 ^a^	0.20 ± 0.01 ^a^	0.27 ± 0.01 ^a^	0.16 ± 0.02 ^a^	0.33 ± 0.02 ^b^
C18:0	ND	ND	ND	0.12 ± 0.01 ^a^	0.09 ± 0.01 ^a^	0.14 ± 0.01 ^a^
C18:1	1.18 ± 0.03 ^a^	1.81 ± 0.28 ^a^	0.86 ± 0.19 ^a^	1.81 ± 0.04 ^b^	1.78 ± 0.29 ^a^	1.83 ± 0.20 ^b^
C18:2	0.32 ± 0.00 ^a^	0.64 ± 0.04 ^a^	0.16 ± 0.02 ^a^	0.39 ± 0.01 ^b^	0.42 ± 0.05 ^b^	0.38 ± 0.03 ^b^
C18:3n-3	ND	ND	ND	0.09 ± 0.01 ^a^	0.08 ± 0.01 ^a^	0.10 ± 0.01 ^a^
∑SFA	98.5 ± 0.14 ^a^	97.55 ± 0.17 ^a^	98.98 ± 0.25 ^a^	97.71 ± 0.15 ^b^	97.72 ± 0.18 ^a^	97.71 ± 0.26 ^b^
∑USFA	1.5 ± 0.01 ^a^	2.45 ± 0.03 ^a^	1.02 ± 0.01 ^a^	2.29 ± 0.02 ^b^	2.28 ± 0.04 ^b^	2.30 ± 0.02 ^b^
∑MCFA	97.55 ± 0.22 ^a^	96.39 ± 0.31 ^a^	98.13 ± 0.19 ^a^	96.33 ± 0.21 ^b^	96.44 ± 0.32 ^a^	96.28 ± 0.20 ^b^
∑LCFA	2.45 ± 0.01 ^a^	3.61 ± 0.02 ^a^	1.87 ± 0.02 ^a^	3.67 ± 0.02 ^b^	3.56 ± 0.03 ^a^	3.73 ± 0.03 ^b^

Data are expressed as mean ± SD; Values with different letters in the same parameter between two CCSKO samples indicate significant differences (*p* < 0.05). HP, CCSKO obtained by hot-pressing method; AEE, CCSKO obtained by aqueous enzyme extraction method; ND, not detected.

**Table 2 foods-12-00293-t002:** Quantitative and qualitative analysis of triglyceride composition of CCSKO.

(**A) Quantitative Analysis of Triglyceride Composition of CCSKO.**			
**Type of Triglyceride**	**C-C-C (ECN = 30)**	**C-C-La (ECN = 32)**	**C-La-La (ECN = 34)**
AEE triglyceride composition (%)	5.40 ± 0.09 ^a^	84.78 ± 3.12 ^a^	9.82 ± 0.30 ^a^
HP triglyceride composition (%)	5.38 ± 0.09 ^a^	84.81 ± 3.59 ^a^	9.71 ± 0.33 ^a^
**(B) Qualitative Analysis of Triglyceride Composition of CCSKO.**			
**Peaks**	**[M + NH_4_]^+^**	**[M + H − RCOOH]^+^**	**Identified Triglyceride Structure**
1	572.4	383.3	C-C-C
2	600.5	383.3, 411.3	C-C-La
3	628.5	383.3, 411.3, 439.3	C-La-La

Data are expressed as mean ± SD; Values with different letters in the same column indicate significant differences (*p* < 0.05). HP, CCSKO obtained by hot-pressing method; AEE, CCSKO obtained by aqueous enzyme extraction method. ECN: Equivalent of carbon number. Abbreviations of fatty acid: C: C10:0; La: C12:0. Abbreviations of fatty acid: C: C10:0; La: C12:0.

**Table 3 foods-12-00293-t003:** Physicochemical properties of CCSKO.

Test Items	Results	
	HP	AEE
Relative density	0.9285 (20 °C/20 °C)	0.9315 (20 °C/20 °C)
Refractive index	1.4491 (20 °C)	1.4532 (20 °C)
Moisture and volatile matter	0.17%	0.13%
Insoluble impurity	0.0059%	0.0063%
Residual solvent content	ND	ND
Acid value	0.88 mg KOH/g	0.40 mg KOH/g
Unsaponifiable matter	0.34%	0.35%
Saponification value	279 mg KOH/g	283 mg KOH/g
Iodine value	4.7 g I_2_/100 g	3.9 g I_2_/100 g
Peroxide value	0.02 mmol/kg	ND
Arsenic	ND	ND
Plumbum	ND	ND
Benzopyrene (α)	ND	ND
Aflatoxins B1	ND	ND

HP, CCSKO obtained by hot-pressing method; AEE, CCSKO obtained by aqueous enzyme extraction method; ND, not detected.

**Table 4 foods-12-00293-t004:** The acute oral toxic effects of CCSKO in SPF ICR mouse.

Gender	Dose Group (g/kg BW)	Number of Mice	Number of Dead Mice	Initial Weight (g)	Final Weight (g)	LD_50_ (g/kg BW)
Male	21.5	5	0	18.08 ± 0.13 ^a^	32.36 ± 1.08 ^A^	≥21.5
	10.0	5	0	18.16 ± 0.21 ^a^	35.52 ± 2.45 ^A^	
	4.64	5	0	18.08 ± 0.08 ^ab^	34.68 ± 2.50 ^AB^	
	2.15	5	0	18.18 ± 0.20 ^a^	36.06 ± 1.02 ^AB^	
(SO)	21.5	5	0	18.17 ± 0.29 ^a^	36.35 ± 1.65 ^C^	
Female	21.5	5	0	18.08 ± 0.08 ^ab^	28.28 ± 1.41 ^B^	≥21.5
	10.0	5	0	18.80 ± 0.89 ^a^	31.06 ± 1.58 ^AB^	
	4.64	5	0	18.14 ± 0.13 ^a^	27.72 ± 0.91 ^B^	
	2.15	5	0	18.06 ± 0.09 ^ab^	30.70 ± 1.94 ^B^	
(SO)	21.5	5	0	18.10 ± 0.26 ^a^	30.96 ± 1.43 ^B^	

Data are expressed as mean ± SD; Values with different letters in the same column indicate significant differences (*p* < 0.05). SO, soybean oil.

**Table 5 foods-12-00293-t005:** The occurrence of micronucleus in mouse PCEs and PCE/NCE value in CCSKO.

Gender	Dose (g/kg BW)	Amount	PCE Observed	PCE with Micronucleus	Rate of Micronucleus (‰)	PCE/NCE
Male	3.44	5	10,035	12	1.20 ± 0.45 ^a^	1.20 ± 0.01 ^a^
	1.72	5	10,029	14	1.40 ± 0.96 ^a^	1.18 ± 0.02 ^a^
	0.86	5	10,043	14	1.39 ± 0.42 ^a^	1.18 ± 0.01 ^a^
	0	5	10,012	12	1.20 ± 0.27 ^a^	1.20 ± 0.03 ^a^
40 mg/kg BW	(CP)	5	10,288	97	9.43 ± 1.22 ^b^	1.11 ± 0.03 ^b^
Female	3.44	5	10,042	15	1.49 ± 0.61 ^a^	1.19 ± 0.02 ^a^
	1.72	5	10,041	19	1.89 ± 0.41 ^a^	1.19 ± 0.01 ^a^
	0.86	5	10,043	16	1.59 ± 0.32 ^a^	1.16 ± 0.03 ^a^
	0	5	10,019	11	1.10 ± 0.42 ^a^	1.17 ± 0.02 ^a^
40 mg/kg BW	(CP)	5	10,072	121	12.01 ± 2.56 ^b^	1.12 ± 0.01 ^b^

Data on the rate of micronucleus and PCE/NCE are expressed as mean ± SD; Values with different letters in the same column indicate significant differences (*p* < 0.05). PCE, polychromatic erythrocytes; NCE, normochromatic erythrocytes.

**Table 6 foods-12-00293-t006:** Result of the bacteria reverse mutation test in CCSKO.

**(A) Results of the samples, unprocessed control and menstruum groups in preliminary bacteria reverse mutation test.**											
**Dose** **(μg/dish)**		**TA97**		**TA98**		**TA100**		**TA102**		**TA1535**	
		**−S9**	**+S9**	**−S9**	**+S9**	**−S9**	**+S9**	**−S9**	**+S9**	**−S9**	**+S9**
Sample	5000	111 ± 8 ^a^	110 ± 19 ^A^	28 ± 2 ^a^	37 ± 8 ^A^	120 ± 4 ^a^	176 ± 16 ^A^	269 ± 35 ^a^	314 ± 34 ^A^	17 ± 2 ^a^	19 ± 1 ^A^
	1582	123 ± 9 ^a^	113 ± 14 ^A^	30 ± 3 ^a^	31 ± 3 ^A^	121 ± 11 ^a^	168 ± 18 ^A^	252 ± 7 ^a^	319 ± 18 ^A^	15 ± 4 ^a^	17 ± 3 ^A^
	501	92 ± 12 ^ab^	113 ± 29 ^A^	29 ± 4 ^a^	41 ± 5 ^AB^	149 ± 15 ^b^	153 ± 16 ^A^	270 ± 20 ^a^	303 ± 25 ^A^	14 ± 3 ^a^	14 ± 2 ^AB^
	158.5	106 ± 13 ^a^	115 ± 19 ^A^	30 ± 4 ^a^	41 ± 5 ^A^	132 ± 14 ^a^	143 ± 3 ^AB^	270 ± 8 ^a^	285 ± 36 ^A^	16 ± 2 ^a^	11 ± 0 ^C^
	50.2	109 ± 25 ^a^	119 ± 27 ^A^	30 ± 4 ^a^	38 ± 4 ^A^	144 ± 28 ^a^	174 ± 11 ^A^	254 ± 26 ^a^	288 ± 14 ^A^	13 ± 1 ^ab^	15 ± 4 ^A^
Unprocessed control		131 ± 5 ^ab^	126 ± 15 ^A^	30 ± 4 ^a^	43 ± 2 ^AB^	143 ± 20 ^a^	158 ± 9 ^A^	285 ± 13 ^a^	277 ± 51 ^A^	16 ± 3 ^a^	16 ± 3 ^A^
Menstruum control		117 ± 20 ^a^	133 ± 29 ^A^	33 ± 2 ^b^	32 ± 2 ^A^	149 ± 17 ^ab^	164 ± 23 ^A^	273 ± 38 ^a^	340 ± 12 ^A^	15 ± 2 ^a^	16 ± 3 ^A^
**(B) Results of the positive groups in preliminary bacteria reverse mutation test.**											
**Positive Control**	**Dose** **(μg/dish)**	**TA97**		**TA98**		**TA100**		**TA102**		**TA1535**	
		**−S9**	**+S9**	**−S9**	**+S9**	**−S9**	**+S9**	**−S9**	**+S9**	**−S9**	**+S9**
NaN_3_	1.5					1321 ± 148 ^c^				986 ± 162 ^c^	
2-AF	10.0		929 ± 93 ^B^		1345 ± 115 ^C^		1387 ± 129 ^C^				
Dexon	50.0	2229 ± 227 ^c^		1049 ± 191 ^c^				1606 ± 161 ^b^			
1,8-DHAQ	50.0								861 ± 111 ^B^		
CP	200										266 ± 43 ^D^
**(C) Results of the samples, unprocessed control and menstruum groups in main bacteria reverse mutation test.**											
**Dose** **(μg/dish)**		**TA97**		**TA98**		**TA100**		**TA102**		**TA1535**	
		**−S9**	**+S9**	**−S9**	**+S9**	**−S9**	**+S9**	**−S9**	**+S9**	**−S9**	**+S9**
Sample	5000	107 ± 18 ^a^	123 ± 21 ^A^	29 ± 3 ^a^	35 ± 1 ^A^	152 ± 8 ^a^	150 ± 9 ^A^	269 ± 10 ^a^	333 ± 8 ^A^	15 ± 1 ^a^	19 ± 1 ^A^
	1000	124 ± 8 ^a^	121 ± 11 ^A^	27 ± 1 ^a^	38 ± 7 ^A^	153 ± 7 ^a^	177 ± 21 ^A^	277 ± 3 ^a^	280 ± 4 ^B^	17 ± 2 ^a^	17 ± 3 ^A^
	200	121 ± 21 ^a^	129 ± 16 ^A^	29 ± 2 ^a^	41 ± 5 ^A^	139 ± 27 ^a^	163 ± 10 ^A^	295 ± 16 ^a^	321 ± 23 ^A^	17 ± 5 ^a^	18 ± 3 ^A^
	40	111 ± 20 ^a^	106 ± 7 ^AB^	30 ± 2 ^a^	37 ± 4 ^A^	145 ± 17 ^a^	175 ± 26 ^A^	254 ± 8 ^b^	296 ± 42 ^A^	15 ± 4 ^a^	17 ± 3 ^A^
	8	99 ± 11 ^a^	115 ± 3 ^A^	33 ± 4 ^a^	32 ± 5 ^A^	130 ± 9 ^ab^	138 ± 15 ^A^	258 ± 24 ^a^	305 ± 20 ^A^	15 ± 4 ^a^	17 ± 3 ^A^
Unprocessed control		95 ± 12 ^a^	105 ± 12 ^A^	32 ± 4 ^a^	43 ± 2 ^B^	156 ± 23 ^a^	148 ± 31 ^A^	284 ± 16 ^a^	285 ± 55 ^A^	12 ± 1 ^ab^	17 ± 4 ^A^
Menstruum control		114 ± 7 ^a^	105 ± 12 ^A^	29 ± 3 ^a^	38 ± 5 ^A^	130 ± 8 ^ab^	131 ± 12 ^A^	275 ± 24 ^a^	281 ± 30 ^AB^	16 ± 3 ^a^	14 ± 2 ^AB^
**(D) Results of the positive groups in main bacteria reverse mutation test.**											
**Positive Control**	**Dose** **(μg/dish)**	**TA97**		**TA98**		**TA100**		**TA102**		**TA1535**	
		**−S9**	**+S9**	**−S9**	**+S9**	**−S9**	**+S9**	**−S9**	**+S9**	**−S9**	**+S9**
NaN_3_	1.5					1117 ± 131 ^c^				906 ± 112 ^c^	
2-AF	10.0		947 ± 80 ^B^		1336 ± 177 ^C^		1178 ± 82 ^B^				
Dexon	50.0	2191 ± 258 ^b^		1090 ± 190 ^b^				1401 ± 151 ^b^			
1,8-DHAQ	50.0								834 ± 79 ^C^		
CP	200										312 ± 46 ^C^

NaN_3_, sodium azide; 2-AF, 2-(2-furyl)-2-(5-nitro-2-furyl) acrylamide; Dexon, p-(dimethylamino) benzenediazo sodium sulfonate; 1,8-DHAQ, 1,8-dihydroxyanthraquinone; CP, cyclophosphamide. Data are expressed as mean ± SD; Values with different letters in the same column indicate significant differences (*p* < 0.05).

## Data Availability

The data are available from the corresponding author.

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
