# Peer review of "Acute Oral Toxicity and Genotoxicity Test and Evaluation of Cinnamomum camphora Seed Kernel Oil"

_foods, 2023, doi:10.3390/foods12020293_

Round 1

Reviewer 1 Report

After reading the article  Acute oral toxicity and genotoxicity test and evaluation of Cinnamomum camphora seed kernel oil  I found the results interesting, however, I have some suggestions for the authors:

1. Rewrite a more concrete conclusion.

2. Attach a section of perspectives

Author Response

Responses to Reviewer #1:

After reading the article “Acute oral toxicity and genotoxicity test and evaluation of Cinnamomum camphora seed kernel oil”, I found the results interesting, however, I have some suggestions for the authors:

1- Rewrite a more concrete conclusion.

Response: We have rewritten the conclusion which is formulated in two aspects: 1) results in physicochemical profiles determination, acute oral toxicity test and a battery of genotoxicity tests; 2) perspectives in usage of CCSKO. Please refer to Lines 530-553.

2- Attach a section of perspectives.

Response: We have attached perspectives in conclusions. Please refer to Lines 550-553.

Reviewer 2 Report

Dear authors

After reviewing the manuscript, my main impression is that the paper titled “Acute oral toxicity and genotoxicity test and evaluation of Cinnamomum camphora seed kernel oil”, is interesting but minor revisions should be done, as indicated in the attached PDF file.

Author Response

Responses to Reviewer #2:

After reviewing the manuscript, my main impression is that the paper titled “Acute oral toxicity and genotoxicity test and evaluation of Cinnamomum camphora seed kernel oil”, is interesting but minor revisions should be done, as indicated in the attached PDF file.

Line 102-104: You have to mention the approved number related to the Institutional Review Board Statement.

Response: We have added the Approval Code in section 2.3. Please refer to Lines 108-109.

Line 105: What is the significance of this acronyme?

If your sentence mean the breed of animals used in your experiments you should write them in italic without abbreviation.

Response: We confirmed that the SPF is for “specific pathogen free” and the ICR is for “Institute of Cancer Research”, and we have added the full names of these two acronyms. Please refer to Line 110.

Line 123: Supresse this one.

Response: We have substituted the preposition “in” to “for” in section 2.4.2. Please refer to Line 128.

Lines 530-532: You cannot confirm the safety of this oil based on single administration (acute toxicity), because the use of essential oils is often repeated and that is why it would have been more interesting if you tested the effect of this oil in a subchronic test.

Response: We have emphasized “preliminary safe” and “no acute oral toxicity” rather than “actual non-toxic” in this manuscript, which allows for a more precise description of the results. Please refer to Lines 21-23, Lines 408-409 and Lines 548-550. In addition, we will further perform the long-term oral toxicity of CCSKO in the future to satisfy the overall safety requirements for the verification of health-functional foods. Please refer to Lines 550-553.

Reviewer 3 Report

Dear Author,

1.      It is mentioned that- CCSKO contains more than 95.00% MCFA and is the only natural MCT found by now in the world…. Kindly revise

2.      Kindly comment on below sentences

a.     However, to our knowledge, CCSKO has not been recognized as a safe oil or new food resource yet.

b.     Due to the minor composition difference between CCSKO and MCT, and CCSKO has been used as a cooking oil since the 1960s (Annals of Anfu County, Jiangxi Province, China)

3.      It is mentioned that- The discovery of edible safety of CCSKO is equivalent to the edible safety discovery of the tomatoes, called the Wolf peach in the late 18th century… kindly comment

4.      It is mentioned that- Strains and culturing in bacterial reverse mutation test (Ames test);  Cell line and culturing in in vitro mammalian cell TK gene mutation test… kindly comment on the subheadings stated as narration does not address any tests conducted … kindly revise as ‘for’ instead of ‘in’

5.      It is mentioned that- The mice in 21.5 g/40 mL groups  were given gavage twice a day… kindly specify which group as soybean and CCSKO both represent the same conc or was it for both the groups

6.      It is mentioned that- The mice were then sacrificed by isoflurane inhalation, and autopsied…. Were they further evaluated . if yes, kindly mention in relation to

7.      It is mentioned that- Under an optical microscope, 2,000 polychromatic erythrocytes (PCE) were counted from each mouse to observe the number of polychromatic erythrocytes containing micronucleus, and the number of mature erythrocytes in the field of 200 polychromatic erythrocytes were observed… kindly verify

8.      It is mentioned that- The experiment should be carried out under the condition of adding or not adding S9…. Kindly revise

Regards

Author Response

Responses to Reviewer #3:

1- It is mentioned that “CCSKO contains more than 95.00% MCFA and is the only natural MCT found by now in the world”. Kindly revise

Response: We have revised this sentence to “CCSKO contains more than 95.00% MCFA and is one of the most important natural MCT found by now in the world”. Please refer to Lines 63-64.

2- Kindly comment on below sentences:

  1. However, to our knowledge, CCSKO has not been recognized as a safe oil or new food resource yet.
  2. Due to the minor composition difference between CCSKO and MCT, and CCSKO has been used as a cooking oil since the 1960s (Annals of Anfu County, Jiangxi Province, China).

Response: a) There are a variety of physiochemical properties in CCSKO such as decreasing body fat deposition and body weight, improving blood lipids profiles, increasing insulin sensitivity and improving dyslipidemia and lipid metabolic disorders. However, there is no reference indicating that CCSKO can be used as an edible oil yet. This study aimed to test and evaluate the acute oral toxicity and genotoxicity of CCSKO as a novel natural MCT oil.

  1. b) We have revised the causal words “due to” in this sentence. Please refer to Lines 75-80.

3- It is mentioned that “The discovery of edible safety of CCSKO is equivalent to the edible safety discovery of the tomatoes, called the Wolf peach in the late 18th century”. Kindly comment.

Response: Tomatoes used to be called the “wolf peach” and considered inedible until a French painter discovered how delicious they were in the late 18th century, accounting for developing the edible value of tomatoes. The edible safety discovery of CCSKO is similar to that of tomatoes. We admit that there is a propaganda tone, but it is helpful to confirm the edible property of CCSKO and make people pay more attention to the development of CCSKO.

4- It is mentioned that “Strains and culturing in bacterial reverse mutation test (Ames test); Cell line and culturing in in vitro mammalian cell TK gene mutation test”. Kindly comment on the subheadings stated as narration does not address any tests conducted. Kindly revise as “for” instead of “in”.

Response: The main role of section 2.4 is to introduce the bacteria strains in Ames test and cell line in in vitro mammalian cell TK gene mutation test, respectively, therefore, we did not mention the preliminary procedures for culturing strains and cell line. We have revised the heading of section 2.4.1 and 2.4.2 to “Strains for bacterial reverse mutation test (Ames test)” and “Cell line for in vitro mammalian cell TK gene mutation test”, respectively. Please refer to Line 120 and Line 128.

5- It is mentioned that “The mice in 21.5 g/40 mL groups were given gavage twice a day”. Kindly specify which group as soybean and CCSKO both represent the same conc or was it for both the groups.

Response: We confirmed that “21.5 g/40 mL groups” represent both CCSKO and soybean oil groups, and we have added the relevant content. Please refer to Line 152.

6- It is mentioned that “The mice were then sacrificed by isoflurane inhalation, and autopsied”. Were they further evaluated. If yes, kindly mention in relation to.

Response: The methods and results of autopsies in mice were shown in section 3.2.2 and section 4.2, respectively. Please refer to Lines 157-159 and Lines 401-403.

7- It is mentioned that “Under an optical microscope, 2,000 polychromatic erythrocytes (PCE) were counted from each mouse to observe the number of polychromatic erythrocytes containing micronucleus, and the number of mature erythrocytes in the field of 200 polychromatic erythrocytes were observed”. Kindly verify.

Response: We confirmed that 2,000 polychromatic erythrocytes (PCE) were counted in order to calculate the PCE/normochromatic erythrocytes (NCE) value, while the number of mature erythrocytes in the field of 200 PCE were observed in order to calculate the micronucleus rate. In addition, we have revised the relevant content in section 3.3.2. Please refer to Lines 178-182.

8- It is mentioned that “The experiment should be carried out under the condition of adding or not adding S9”. Kindly revise.

Response: We have revised this sentence to “The experiment was carried out under the condition of adding and not adding S9”. Please refer to Lines 223-224.

Round 2

Reviewer 1 Report

The authors responded to the suggestions made.